# *Cryptococcus neoformans* Slu7 ensures nuclear positioning during mitotic progression through RNA splicing

**Vishnu Priya Krishnan**, **Manendra Singh Negi**, **Raghavaram Peesapati**, **Usha Vijayraghavan***

Department of Microbiology and Cell Biology, Indian Institute of Science, Bangalore, India

* uvr@iisc.ac.in

**Data Availability Statement:** All relevant data are within the paper and its Supporting Information files.

## Abstract

The position of the nucleus before it divides during mitosis is variable in different budding yeasts. Studies in the pathogenic intron-rich fungus *Cryptococcus neoformans* reveal that the nucleus moves entirely into the daughter bud before its division. Here, we report functions of a zinc finger motif containing spliceosome protein *C. neoformans* Slu7 (CnSlu7) in cell cycle progression. The budding yeast and fission yeast homologs of Slu7 have predominant roles for intron 3' splice site definition during pre-mRNA splicing. Using a conditional knockdown strategy, we show CnSlu7 is an essential factor for viability and is required for efficient cell cycle progression with major role during mitosis. Aberrant nuclear migration, including improper positioning of the nucleus as well as the spindle, were frequently observed in cells depleted of CnSlu7. However, cell cycle delays observed due to Slu7 depletion did not activate the Mad2-dependent spindle assembly checkpoint (SAC). Mining of the global transcriptome changes in the Slu7 knockdown strain identified downregulation of transcripts encoding several cell cycle regulators and cytoskeletal factors for nuclear migration, and the splicing of specific introns of these genes was CnSlu7 dependent. To test the importance of splicing activity of CnSlu7 on nuclear migration, we complemented Slu7 knockdown cells with an intron less *PAC1* minigene and demonstrated that the nuclear migration defects were significantly rescued. These findings show that CnSlu7 regulates the functions of diverse cell cycle regulators and cytoskeletal components, ensuring timely cell cycle transitions and nuclear division during mitosis.

## Author summary

Nuclear position in eukaryotic cells is spatio-temporally regulated during the cell cycle. Unlike in *Saccharomyces cerevisiae*, where nuclear migration to mother-daughter bud neck precedes mitotic segregation, in *Cryptococcus neoformans*, the nucleus moves entirely into the daughter bud before division. Transcription dynamics during the *C. neoformans* cell cycle show periodic expression changes and since all nascent pre-mRNAs must be processed to mRNA, splicing is indispensable for gene expression. Yet, how

**Funding:** This work was supported by Institute of Eminence fund (IE/REDA-21-1385, IE/REDA-22-1385, IE/REDA-23-1385) to UV. The work is also supported by the fellowship from Department of Biotechnology Junior Research fellowship program (DBT/2016/IISc/716) to VP. MSN and RP were supported by fellowship from Indian Institute of Science. The funders had no role in study design, data collection and analysis, decision to publish, or preparation of the manuscript.

**Competing interests:** The authors have declared that no competing interests exist.

evolutionarily conserved splicing factors modulate the complex intron-rich *C. neoformans* transcriptome and their impacts on atypical mitotic nuclear position is unexplored. Here, we show CnSlu7 a zinc finger motif spliceosome factor is essential for viability and for cell cycle progression with a major mitotic role. Live imaging of the nucleus, the spindle and other factors revealed significant abnormalities in nuclear migration from mother to daughter cells, as well as nuclear and spindle position defects upon CnSlu7 depletion. Depletion of CnSlu7 altered the transcript status of various cell cycle players, including those critical for nuclear migration, as their introns were CnSlu7 dependent for splicing. Together, these findings highlight the roles of *C. neoformans* Slu7 in fine-tuning gene expression levels of transcripts that ensures timely cell cycle progression and spatial control of nuclear position.

## Introduction

The position of the nucleus during symmetric or asymmetric cell division is spatio-temporally regulated and coordinated with cell cycle progression in eukaryotes [1,2]. In budding yeast *Saccharomyces cerevisiae*, with cell division resulting in two cells of asymmetric cell size, nuclear migration to the neck between the mother and daughter cell is essential for the subsequent segregation of chromosomes [3]. In the case of fission yeast, *Schizosaccharomyces pombe*, the positioning of the nucleus at the center determines the position of the cell division plane, guaranteeing each daughter cell inherits one copy of the genome [4]. The nuclei migrate towards the growing hyphal tip in the filamentous fungus *Aspergillus nidulans*, where nuclei are anchored in the hyphal compartments and the mitotic exit is followed by septation [5]. In contrast, recent studies in basidiomycete fungi such as *C. neoformans* and *Ustilago maydis* show that during mitosis, nuclei move entirely into the daughter cell, where chromosome segregation and spindle elongation is initiated [6,7]. These striking differences in the nuclear position in different fungal species, with closed mitosis or semi-open mitosis, can arise from variations in interactions between the nucleus and actin and microtubule cytoskeleton that are mediated by the molecular motors and nuclear envelop proteins. However, the molecular and biophysical mechanisms underlying the less common pattern of nuclear migration to daughter bud in basidiomycete yeasts remain unclear.

In *S. cerevisiae* and *S. pombe*, nuclear movement during cell division is mediated by pushing and pulling forces generated by microtubules (MTs) and associated motor proteins such as kinesin1 and dynein [8,9]. Genetic studies in several fungal species identified dynein as a critical determinant for the positioning of the nucleus and the spindle [10], with LIS1/PacI and NudE being regulators that influence the spatial distribution of dynein [11,12]. Recent biochemical and cryo–EM studies have reported the role of LIS1/Pac1 in dynein regulation in *S. cerevisiae* [13–15]. Studies in *C. neoformans* have demonstrated the mechanics of nuclear migration and spindle positioning by time–lapse live cell imaging and quantitative modelling [6,16,17]. Transcriptional profiling of *C. neoformans* cell cycle revealed periodic changes in the expression of 1134 out of 6182 transcripts tested [18]. Periodic expression of cell cycle proteins is also known in human and budding yeast cells, and these gene expression profiles are regulated at transcriptional, post-transcriptional, and post-translational levels [19–22]. Since all newly transcribed pre-mRNA must be processed to form mRNA that yields functional proteins, splicing is an indispensable contributor to the fine-tuning of gene expression for cell cycle progression.

Numerous studies in model yeasts and human cell lines have shown the interlinking between pre-mRNA splicing and cell cycle progression [23]. Changes in the expression levels or activity of splicing factors can directly affect the expression of specific transcripts encoding proteins that play critical roles in the cell cycle. Evidence that strongly supports these assumptions comes from genetic screens for regulators of the cell division cycle identified mutants in core pre-mRNA splicing factors of the megadalton spliceosomal complex [24–27]. In fungi, the biological impact of splicing on cellular processes comes from studies done in *S. cerevisiae* and *S. pombe*, genomes where <5% and 43% of genes, respectively have introns [28,29]. The splicing dependent roles for budding and fission yeast spliceosomal factors ScPrp17, SpPrp18, and SpPrp16 in cell cycle [30–32] have been reported. Although these reports establish splicing as one important mechanism in the functional coordination of gene expression, cell cycle progression, and cell survival in budding yeast *S. cerevisiae* and *S. pombe*, its role in basidiomycete fungi such as *C. neoformans* and *U. maydis* with atypical features during cell division is not fully understood.

The human pathogenic yeast *C. neoformans* is a genetically tractable model system for investigating the functional connections between RNA splicing and cellular processes that underlie nuclear migration and segregation in the daughter cell. The complex transcriptome with introns in more than 99% of its genes hints towards the likely roles of introns in gene expression and genome stability [33,34] and plausible roles of splicing factors in the regulation of cell cycle transcripts. *C. neoformans* undergoes semi-open mitosis in which the nuclear envelop partially ruptures during the chromosome segregation in the daughter bud [6]. In this study, we have identified how the splicing factor CnSlu7 impacts mitotic progression during *C. neoformans* cell cycle. Through time–lapse live imaging of cells with conditional knockdown of CnSlu7, we demonstrate that Slu7 is required for proper nuclear and spindle positioning during mitosis. Depletion of Slu7 affected multiple players in cell cycle, suggesting splicing mediated fine tuning of expression during different phases of cell cycle progression. Here, we elucidate the molecular mechanisms through which CnSlu7 controls nuclear position prior to nuclear division by promoting efficient splicing of *PAC1* transcript whose protein ortholog in *S. cerevisiae* is involved in nuclear migration during cell division.

## Results

### Knockdown of Slu7, a zinc knuckle motif splicing factor in *Cryptococcus neoformans* leads to slower G2–M progression

Previous studies of *slu7* mutants in *S. cerevisiae* and the conditional knockdown of its *S. pombe* ortholog have reported this spliceosome factor is encoded by an essential gene in both these model fungi [35,36]. The predicted *C. neoformans* Slu7 protein (CnSlu7, *CNAG_01159*) has the CCHC type conserved zinc finger motif seen in all its eukaryotic homologs, yet this protein is the most divergent among the fungal homologs and is closer to its human counterpart (**Figs 1A and S1A**). The multiple sequence alignment revealed that the N terminal region is highly conserved while the C terminal region of Slu7 protein is diverged among the eukaryotes. To gain functional insight on CnSlu7, we generated the strain *GAL7p-SLU7* (*slu7kd*) for conditional knockdown where the chromosomal copy was replaced with an expression cassette for mCherry-tagged Slu7 and configured for expression from the galactose-inducible *GAL7* promoter (**S1B Fig**). Growth of *GAL7p-SLU7* (*slu7kd*) on glucose-containing media that represses expression from *GAL7* promoter (non-permissive condition) was severely retarded (**Fig 1B**), contrasting the robust growth on galactose containing media (permissive conditions). To assess Slu7 protein levels and RNA levels, cultures grown in glucose media for 6 hrs (hours) or 12 hrs were taken for western blot analysis and RT-PCR assays. By 12 hrs in repressive growth

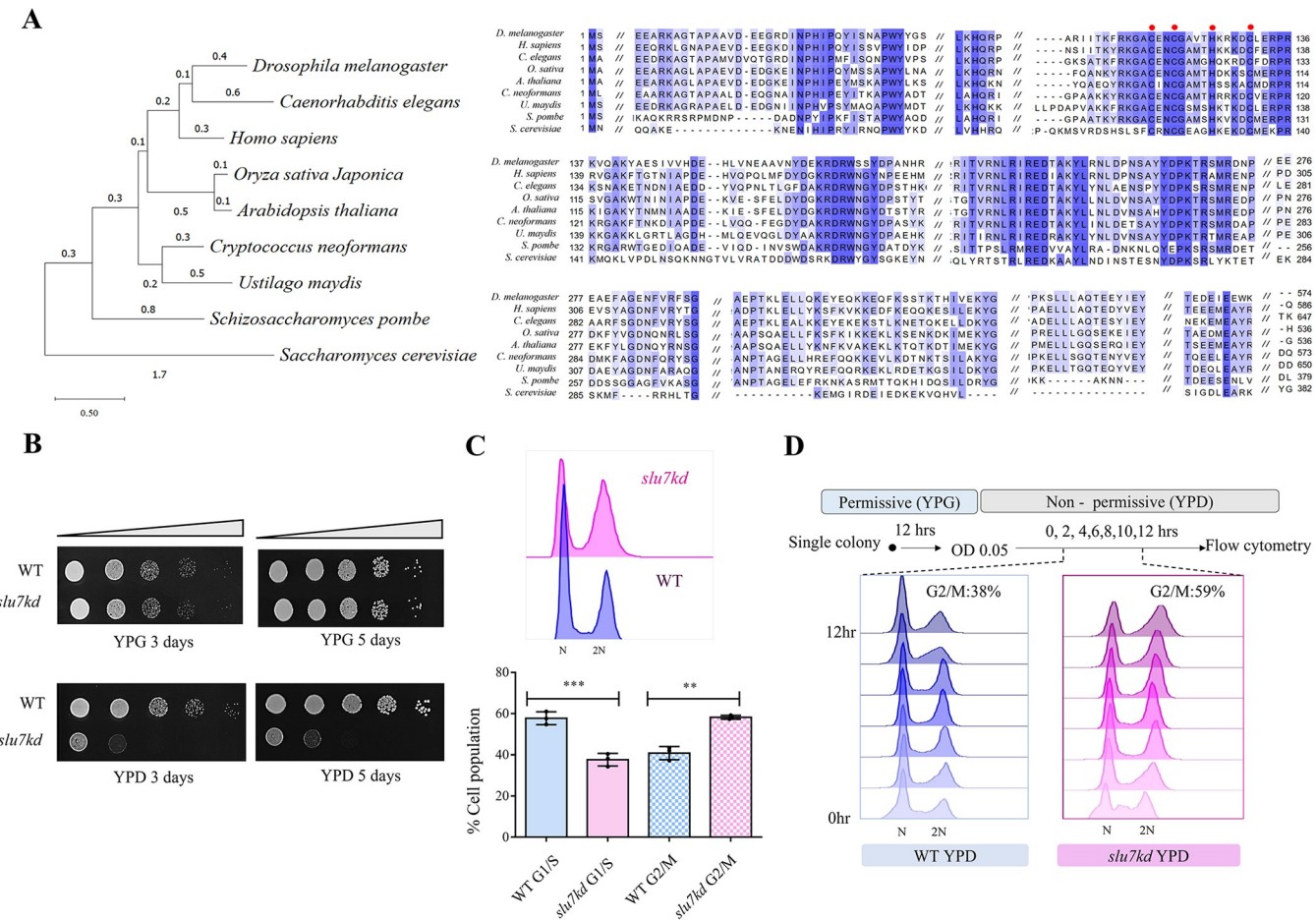

**Fig 1. Slu7 is a highly conserved C2HC zinc finger protein required for cell cycle progression during mitosis.** (**A**) Phylogenetic tree conservation of Slu7 protein among eukaryotes. The tree was generated using MEGA 11 with Maximum Likelihood method. The branch lengths measured in the number of substitutions per site is denoted above the branches. The red dots denote the conserved zinc finger knuckle motif. (**B**) Serial 10-fold dilution starting with 2 X $10^5$ cells from wildtype and *slu7kd* strain was done, followed by plating on permissive (YPG) and non-permissive media (YPD). Growth after incubation at 30°C for 3 days and after 5 days is shown. (**C**) Flow cytometry analysis of cells from wildtype and *slu7kd* cultures grown in non-permissive media (YPD) for 12 hours. The bar graph data represents mean ± SD for three independent biological replicates. (**D**) Flow cytometry analysis of cells from wildtype and *slu7kd* strain withdrawn at various time points inoculation of early log phase cells into non-permissive media (every two hours). The percentage figures given at the top represent the % of cells in the G2/M phase at the end of 12 hours, N = 3.

media, *SLU7* transcript level was highly reduced, and protein was undetectable, in line with the observation of severe growth arrest (**S1C and S1D Fig**). Thus, these data affirm that *SLU7* is an essential gene in *C. neoformans*.

Previous studies in fission yeast have shown that splicing of longer introns depends on SpSlu7 [36]. We performed semi-quantitative RT-PCR (reverse transcription polymerase chain reaction) analysis to assess the splicing role of the predicted *C. neoformans* Slu7 homolog. We first assessed *TFIIA intron 1* (187 nucleotides [nts] in length) for its splicing as it was a long intron compared to the average intron length of 56 nts in this genome. The splicing of *intron 1* of *TFIIA* was impaired as unspliced E1-I1-E2-E3 are detected at higher levels in the *slu7kd* when compared to wildtype. Also detected at low levels are unspliced precursors representing E1-I1-E2-I2-E3, indicating compromised splicing of even the 96 nts *TFIIA intron* 2. These data confirm the conserved function of Slu7 as a splicing factor (**S2A Fig**).

The growth kinetics and progression through cell division of the *slu7kd* strain after transfer to non-permissive glucose media (YPD) was monitored. In comparison with similarly treated

wildtype (WT) cells, the growth kinetics of *slu7kd* was marginally slow even in early time points (2–3 hrs), and this defect was noticeably enhanced by 6 hrs with a 2-fold difference in culture O.D. by 10 hrs in non-permissive media (**S2B Fig**). Flow cytometry of these cell populations was taken to determine the terminal cell cycle phenotype after 12 hrs in YPD. In *slu7kd* cultures grown in glucose, an increase in the population of cells in the G2/M phase was detected, which contrasts with WT cells having a higher percentage of cells in the G1 phase when grown in the same media (**Fig 1C**). In culture aliquots withdrawn every 2 hrs after transfer from permissive to non-permissive media (**Fig 1D**), we noted an increase in the G2/M population in the *slu7kd* strain with significant effects from 6 hrs and beyond. Flow cytometry was done with *slu7kd*, and wildtype cells synchronized in the late G1/early S phase by the addition of 15mg/ml HU. After 6 hrs of growth in non-permissive media (initial depletion + arrest), these cells were released into fresh non-permissive media. We observed that the after release from HU arrest cells with knockdown of CnSlu7 had an overall increase in the G2/M population compared to the congenic wildtype strain, which efficiently resumed the cell division cycle (**S2C Fig**). In addition, *slu7kd* cells resumed growth without losing viability when shifted from non-permissive to permissive conditions, indicating the arrest phenotype associated with the depletion of Slu7 is largely reversible and does not cause chromosome mis-segregation (**S2D Fig**). Taking these results together, we conclude that the inability of the conditional mutant to grow in the non-permissive media is due to impaired progression through the G2–M phase of the cell cycle.

## Knockdown of Slu7 leads to nuclear mispositioning during mitosis

For an in-depth understanding of the role of Slu7 during the G2 and M phases of the *C. neoformans* cell division cycle, we generated strains with Slu7 conditional knockdown and one of several fluorescently tagged markers. These strains would allow tracking of nuclear position/its migration (GFP-H4), kinetochore clustering (GFP-CENPA), and spindle dynamics (GFP-TUB1) in live cells under conditions where Slu7 is either abundant or strongly depleted. The slow growth in YPD non-permissive media for these *slu7kd* marked strains, with different vital readouts for mitotic progression, was similar to the growth defects seen in the parent *slu7kd* (**S3A Fig**). We performed live imaging of nuclei in *slu7kd* GFP-H4 cells after 12 hrs of growth in non-permissive media and compared these data with live imaging of nuclei in wildtype cells taken as a control (GFP-H4 alone). Tracking the nucleus in wildtype cells with small bud showed that the nucleus moved into the daughter bud, divided into two, and half of the nuclear mass moves back into the mother cell, and the other half is retained in the daughter bud (**S3B Fig**, top row).

In contrast, in *slu7kd* GFP-H4 cells, no nuclear movement was observed under conditions where Slu7 was depleted, and we noted an increased population of cells with unsegregated nuclei (**S3B Fig**, bottom row). We measured large budded cell population for proportion with either segregated nuclei or unsegregated nuclei by assessing nuclei in fixed *slu7kd* cells or wildtype cells, each grown in non-permissive media for 6 hrs and 12 hrs. We note *slu7kd* cultures, as compared to the wildtype, had a higher proportion of large budded cells with either segregated or unsegregated nuclei, and these changes were evident even within 6 hrs of growth in non-permissive media (**Fig 2A**, yellow and red bar graphs). Based on these data, for further analysis of cell cycle and mitosis, cultures of *slu7kd* were imaged after growth for 6 hrs in glucose media and compared with wildtype cultures raised in parallel. This would reveal the temporally early consequences of Slu7 depletion.

Live imaging of the nucleus in GFP-H4 wildtype cells showed that with cell cycle progression, the undivided nucleus moves from the mother entirely into the daughter cell/bud.

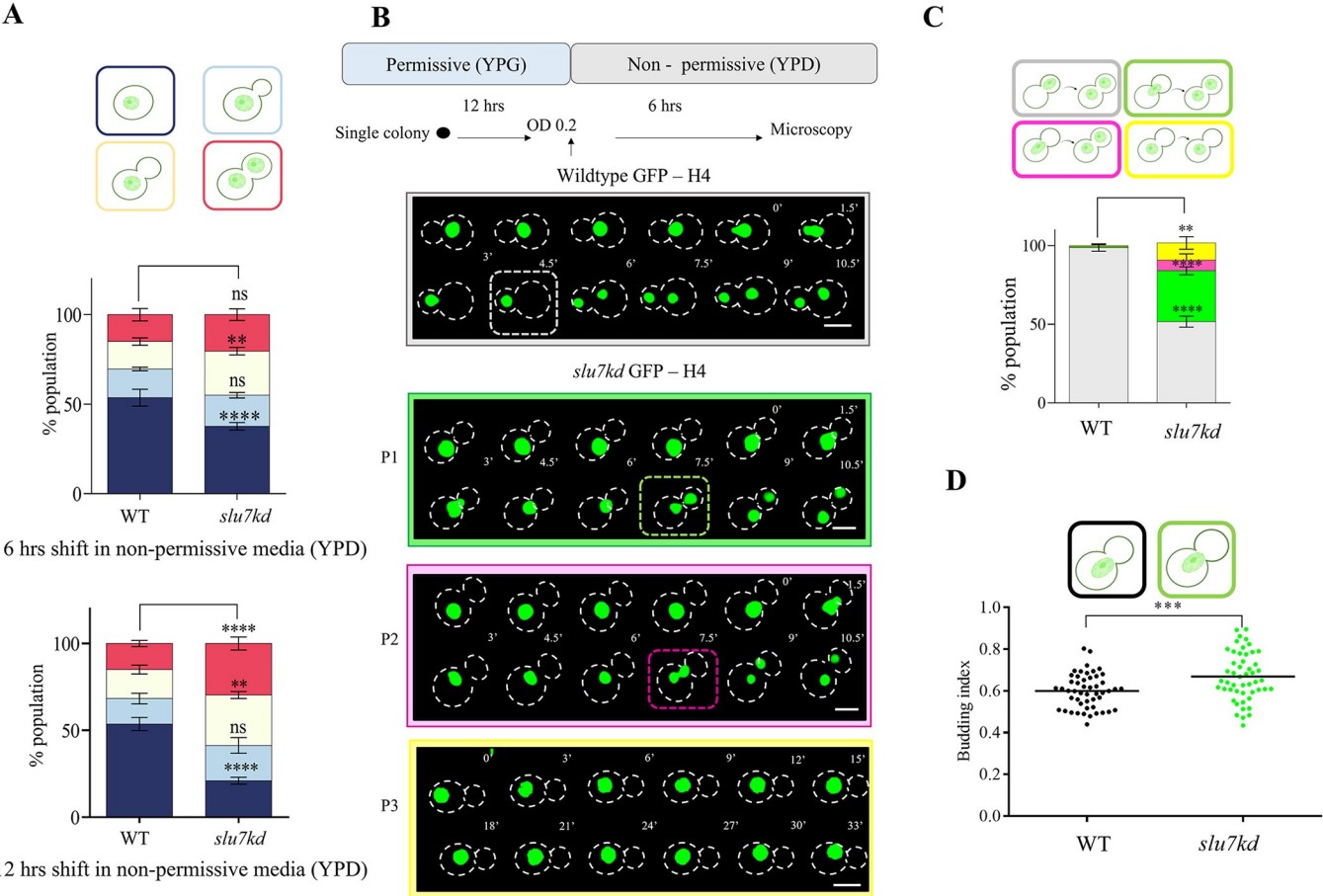

**Fig 2. Depletion of Slu7 results in nuclear mispositioning during G2 to M transition.** (**A**) The percentage of cells at various phases of the cell cycle was measured using *slu7kd* (n = 100) and wildtype (n = 100) in the background of GFP-H4 (fixed cell with 4% paraformaldehyde) grown in YPD for 6hrs and 12hrs, respectively. Cells were scored based on the budding index and the position of the nucleus. The data represent mean ± SD for three independent biological replicates. One-way ANOVA test followed by Turkey's multiple comparison test was used to calculate the statistical significance of differences between the population (the p values show the difference compared to the wildtype). (**B**) Time-lapse snapshots of *slu7kd* and wildtype cells with GFP-H4 reporter to visualize nuclear dynamics after growth in non-permissive media for 6 hours. T = 0 was taken when the nucleus enters the neck region. Bar, 5μm. (**C**) Quantification of defects in the nuclear migration in the wildtype and *slu7kd* carrying histone GFP-H4. Percentages of cells showing proper nuclear migration into daughter bud in large budded cells followed by division (wildtype phenotype), nuclear migration defect with nuclei being divided at the neck between the mother and daughter bud (P1), nuclear migration defect with nuclei being divided in the mother cell (P2) and no nuclear migration (P3) are plotted. The data represent mean ± SD for three independent biological replicates with 50 cells each. One-way ANOVA test followed by Turkey's multiple comparison test was used to calculate the statistical significance of differences between the population (the p values show the difference compared to wildtype). (**D**) Analysis of budded cells with an unsegregated nucleus in the wildtype and *slu7kd* cells expressing histone GFP-H4. Bar, 5μm. The budding index was measured when the nucleus reached the neck. Unpaired t-test was used to calculate the statistical significance of differences between the population (the p values show the difference compared to wildtype), n = 50.

Nuclear division initiates in the bud, and one of the divided nuclei with segregated chromosomes move back to the mother cell (**Fig 2B**, top panel, wildtype, quantified in **Fig 2C** grey bar). In these wildtype cells, the approximate time taken from positioning the nucleus at the bud neck to segregation of the divided nuclei in the mother and daughter cell is about 6–7 mins. Strikingly, in *slu7kd* cells depleted for Slu7 by growth in repressive media for 6 hrs, we observe GFP-H4 labelled nuclei have some distinct phenotypes: a) nuclear division in daughter cell that is similar in temporal and spatial pattern to that in the wildtype, b) abnormal nuclear division at the neck between the mother and daughter cell (**Fig 2B** green framed panel labelled P1, quantified in **Fig 2C** green bar), c) abnormal nuclear division in the mother cell (**Fig 2B**

pink framed panel labelled P2, quantified in **Fig 2C** pink bar), d) nucleus remains unsegregated in mother cell (**Fig 2B** yellow framed panel labelled P3, quantified in **Fig 2C** yellow bar). These data were quantified for several cells imaged (**Fig 2C**). In addition, we observed a striking difference in the budding index of wildtype cells at the stage when the nucleus is positioned at the bud neck as compared to the budding index quantified for *slu7kd* cells with nucleus at the mother-daughter neck (**Fig 2D**). This indicates that the bud growth is uncoupled from mitotic progression in *slu7kd* cells.

In ascomycete yeasts such as *S. cerevisiae* and *C. albicans*, kinetochores are clustered from the G1 phase of the cell cycle and are attached to the spindle pole body and the microtubules [37,38]. In contrast, in the basidiomycete *C. neoformans*, kinetochores are unclustered in G1 through interphase and clustered on the transition to mitosis. This clustering and kinetochore association with microtubules lead to nuclear migration in the daughter bud [6]. Since we had observed nuclear migration from the mother to the bud was significantly perturbed even after 6 hrs of growth of *slu7kd* cells in non-permissive conditions, live-cell imaging was done to investigate spatiotemporal dynamics of the inner kinetochore protein CENPA in wildtype and *slu7kd* cells. These experiments with GFP-CENPA marked cells mirrored the phenotype seen in GFP-H4 marked cells (wild type and *slu7kd*, **S3C Fig**). Together, these results confirmed that without affecting the kinetochore clustering, depletion of Slu7 affects nuclear migration during the G2 to mitotic transition in *C. neoformans*. Guided by these observations, our subsequent studies investigated possible underlying mechanisms by which CnSlu7 could regulate nuclear migration and cell division.

The knockdown of Slu7 resulted in increased sensitivity to Thiabendazole (TBZ), a microtubule depolymerizing drug, even when used at a sub-lethal 4μg/ml concentration (**S4A Fig**). Growth kinetics of the wildtype strain was unperturbed in the same conditions. We tested the consequences of adding sub-lethal concentrations of TBZ to cultures that were allowed to resume the cell cycle after Hydroxyurea (HU)-based synchronization in the early S phase. The earliest notable defects seen in *slu7kd* cells were at 2 hrs and 3 hrs after the release from HU arrest and resumption of cell cycle (**S4A Fig**). The hypersensitivity to TBZ and the slow accumulation of cells with 2N DNA content suggest a disruption in microtubule mediated and microtubule associated processes during cell cycle in Slu7 depleted cells. These results hint at the dependency on Slu7 for normal expression levels for factors associated with microtubules.

To investigate whether Slu7 knockdown triggers the activation of spindle assembly checkpoint (SAC), we generated a strain with conditional *slu7kd* in cells with *mad2Δ* allele and the GFP-H4 nuclear marker. The growth kinetics of the double mutant *mad2Δ slu7kd* was not significantly different from that of *slu7kd* however, a subtle synthetic sick phenotype was seen (**S4B Fig**, compare colony growth in row 2 and row 4). Further, flow cytometry analysis of HU synchronised *mad2Δ slu7kd* cells released for progression through cell cycle showed a similar percentage of cells in G2/M phase as the parent *slu7kd* (**S4C Fig**). Live imaging of the nuclear migration revealed no exacerbation or rescue of this defect in the *mad2Δ slu7kd* double mutant compared to the *slu7kd* cells (**S4D Fig**). The abrogation of the SAC did not override the G2/M arrest phenotype of *slu7kd*. Thus, we identify that the depletion of CnSlu7 severely affects the temporal and spatial sequence of events during mitosis, particularly nuclear migration and division.

## Mitotic spindle is mis localized on depletion of Slu7

In *S. cerevisiae*, cytoplasmic microtubules (cMTs) govern the nuclear migration from the mother cell to the bud neck through motor functions of kinesins, dynein, and linking proteins [8,39]. Dynamic growth and shrinkage kinetics of cMTs and the force exerted by the plus and

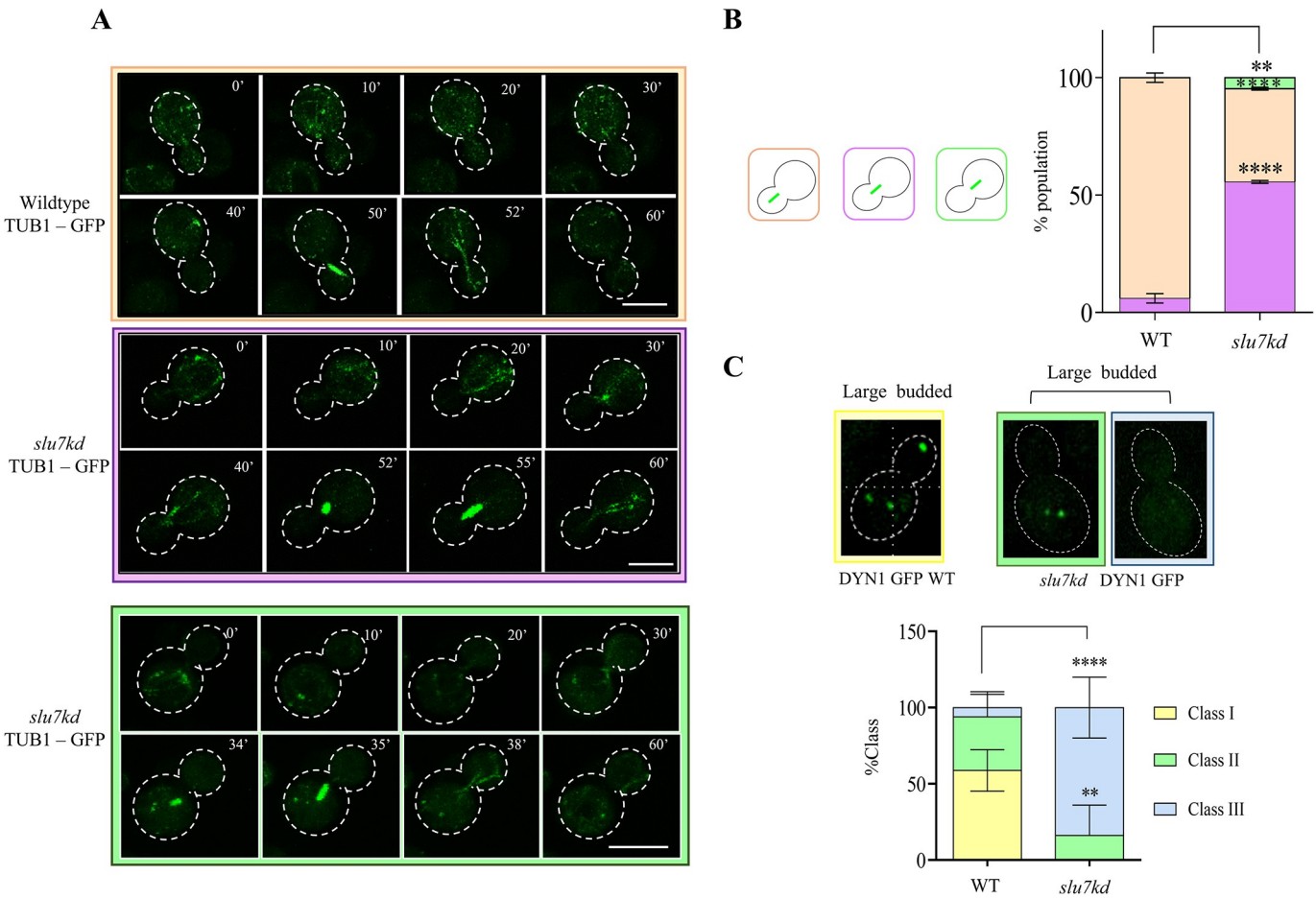

**Fig 3. Knockdown of Slu7 leads to mitotic spindle mis-localization and loss of localized dynein puncta in daughter cell.** (**A**) Time-lapse snapshots of *slu7kd* and wildtype cells with GFP-TUB1 to visualize the spindle dynamics when the strains are grown in YPD (non-permissive for *slu7kd*) media for 6 hours. Bar, 5μm. (**B**) The mitotic spindle position in the wildtype and *slu7kd* cells both marked with GFP-TUB1 is schematically represented and quantified in both genotypes. The data represent mean ± SD for three independent biological replicates with 50 cells each. One-way ANOVA test followed by Turkey's multiple comparison test was used to calculate the statistical significance of differences between the population (the p values show the difference compared to wildtype). (**C**) Localization of Dyn1 in the wildtype and *slu7kd* cells with Dyn1-3xGFP after growth in the non-permissive conditions. Percentages of cells with different patterns of dynein signal are quantitatively represented in the bar graph. The yellow bar represents cells with clustered dynein puncta both in mother and daughter bud in large budded cells. The green bar represents cells with multiple dynein puncta only in the mother bud of large-budded cells, and blue bar shows % cells with no dynein puncta either the mother and daughter bud of large budded cells. The data represent mean ± SD for three independent biological replicates with n ≥ 65 large-budded cells. Bar, 5μm.

minus end-directed motor proteins such as dynein and kinesin cause nuclear migration [8]. Here, we visualized cytoplasmic microtubules in *C. neoformans* strains using GFP-TUB1 tagged wildtype and *slu7kd* GFP-TUB1 strains. Both strains after 6 hrs in YPD, were taken for live imaging. With cell cycle progression from G2 to M in wildtype cells, the cMTs organize into a unipolar MT cytoskeleton with plus-end/ one-end penetrating the growing bud (**Fig 3A**, wildtype panel, time point 50 mins).

In *slu7kd* cells, in cells with small bud, numerous cMTs were nucleated from the MTOCs, and as the cell cycle progressed, they organized to form the mitotic spindle (**Fig 3A**, *slu7kd* GFP-TUB1 panel, time point 55 mins). However, in cells with the mitotic spindle, indicative of metaphase, the spindle was positioned in the neck of the mother. This contrasts with the normal mitotic spindle position entirely in the daughter cell in the wildtype strain (**Fig 3A** wildtype GFP-TUB1 panel, time point 50 mins time point to be compared with *slu7kd* panel 55

mins time point). These data were quantified for images from several cells, and we observed that 45% of cells in *slu7kd* strain showed spindle mislocalized to the neck (**Fig 3B**, purple bar), and in 5% of the population, the mitotic spindle was entirely localised in the mother cell (**Fig 3B**, green bar). A similar phenotype of spindle mispositioning was observed in *S. cerevisiae* mutants in *pac1Δ* and *dyn1* [11]. This defect was linked to the lack of force from the microtubule on the nucleus provided by the motor protein dynein. Taking this lead, we generated *slu7kd* allele in DYN1-3XGFP strain. In these marked strains, we studied the spatial distribution of dynein puncta in mother and daughter cells to understand the consequences of conditional depletion of CnSlu7. Wildtype and *slu7kd* cells with DYN1-3XGFP were imaged after 12 hrs of growth in YPD media. In large budded wildtype cells, we observe two patterns for dynein distribution. Firstly, cells with multiple dynein puncta only in the mother cell and no dynein signal in the daughter cell. Secondly, cells display a bright dynein punctum in the daughter bud and multiple dynein puncta in the mother cell (**Fig 3C**, wildtype framed image panel and yellow bar). No signal was detected in a very small percentage of wildtype cells (**Fig 3C**, WT blue bar). In contrast, *slu7kd* cells had an overall loss of dynein signal, with a significant proportion of cells displaying no dynein puncta in either mother or daughter cells (**Fig 3C**, *slu7kd* blue framed image, and blue bar). Few *slu7kd* cells had dynein puncta only in the mother cell (**Fig 3C**, *slu7kd* green framed image, and green bar) at a proportion much lower than in wildtype cells (**Fig 3C**, WT green bar). The abnormal dynein distribution seen on CnSlu7 depletion is reminiscent of dynein phenotype in the *S. cerevisiae pac1Δ* mutant [11].

## CnSlu7 regulates the expression levels of a subset of cell cycle genes

To understand the genome wide impact on gene expression triggered by Slu7 depletion, the transcriptome in *slu7kd* cells grown in non-permissive media for 12 hrs was compared to that of wildtype cells grown in parallel. Analysis of paired-end RNA-Seq data from three biological replicate datasets for each strain yielded genes whose expression levels were affected with stringent cut-off parameters (Materials and Methods, q value $\leq$ 0.05 and foldchange > 2; **S5A Fig**, **S3 Table**). Of the 1389 deregulated genes, 808 genes had lower transcript levels in *slu7kd* cells, while transcript levels were higher for 581 genes (**S5B Fig**). Transcripts coding for proteins of cell cycle regulators, transporters, transcriptional factors, virulence factors, and genes involved in anabolism predominated the set with lower levels in *slu7kd*. The most highly enriched functional categories are related to integral components of membranes, cell cycle, and translation (**Fig 4A**). Since we have observed that *slu7kd* cells after 12 hrs in non-permissive YPD media accumulate at G2 phase and have impaired nuclear migration during mitosis, we curated from the list of genes deregulated in *slu7kd* cells a subset of genes with predicted roles in *C. neoformans* cell cycle and nuclear migration based on their homology to their *S. cerevisiae* counterparts (**Fig 4B**) [10,11,17,40]. A heat map depicting the expression levels of the genes in the curated list revealed that most of these genes were deregulated in *slu7kd* when compared to wildtype. These deregulated cell cycle genes included *CLN1* and *PAC1*, both with significantly reduced low transcript levels in *slu7kd* cells (6-fold and 4-fold, respectively), as also validated by qRT-PCR (**Fig 4C**). We also investigated the status of other transcripts encoding functions related to cell cycle, such as *ROM1*, *PCL1*, *BIM1*, *DAD4*, *PLK1*, and observed these transcripts were also deregulated on depletion of Slu7 (**Fig 4C**).

## Slu7 is required for efficient splicing of *PAC1* transcript with implications for nuclear migration

In *S. cerevisiae* Pac1 is one of the key players that ensures nuclear migration during mitosis [11] as it targets dynein to the plus end of microtubules [13–15]. Hence, we investigated the

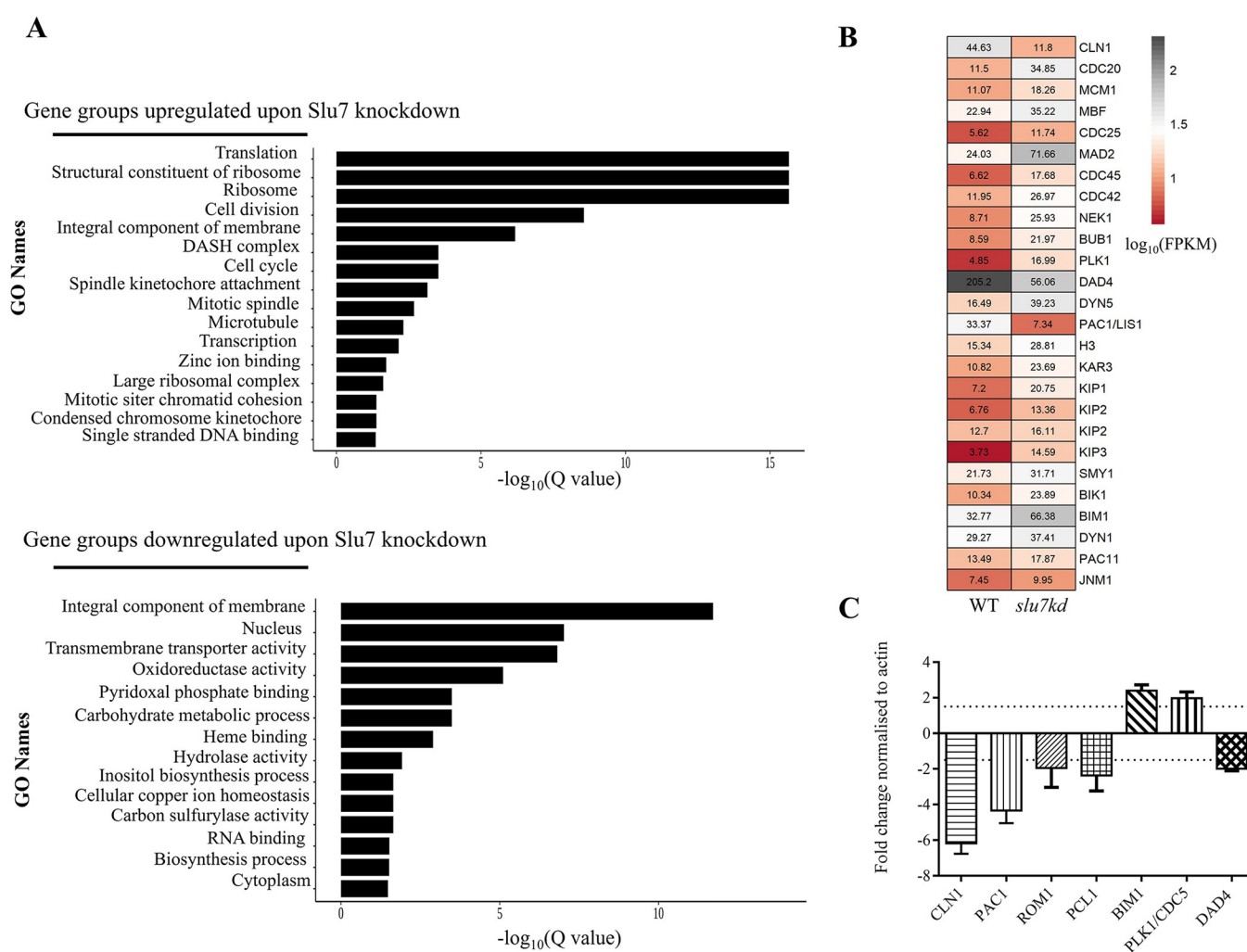

**Fig 4. Slu7 knockdown leads to downregulation of transcripts involved in diverse roles in cell cycle.** (**A**) Analysis of genes that are significantly upregulated and downregulated after Slu7 knockdown using GO analysis. (**B**) Heatmap representing the curated list of cell cycle-related genes deregulated upon depletion of Slu7. The scale represents log10 (FPKM) obtained for genes in wildtype and Slu7 knockdown. (**C**) Assessing the levels of RNA downregulation for a subset of genes involved in cell cycle in knockdown and wildtype cells after the shift into non-permissive media for 12 hours by qRT-PCR.

splicing status of *C. neoformans PAC1* (*CNAG_07440*) and compared these data between wild-type *vs. slu7kd* cells after growth for 12 hrs in YPD media. *PAC1* gene in *C. neoformans* contains six introns, with an average intron length of about 50 nucleotides (nts), similar to the global intron average for this species. We hypothesized that one or more of these introns could be splicing substrates of CnSlu7. Semi-quantitative RT-PCR assays were performed to measure the pre-mRNA and mRNA levels for each exon-intron-exon junction in the *PAC1* transcript. The fold reduction in each spliced mRNA junction and the change levels of each unspliced precursor mRNA were determined after Slu7 depletion for 12 hrs. We observed that *PAC1* intron 1 is poorly spliced in the *slu7kd* cells, with a significant reduction in the amplicon representing the spliced mRNA across this junction and an increase in unspliced E1-I1-E2 levels as compared to wildtype cells (**Fig 5A**). The splicing of *PAC1* intron 2 was also compromised on Slu7 depletion by conditional knockdown, as reflected by mild pre-mRNA accumulation and a decrease in mRNA, as compared to wildtype cells, but the splicing defect was not as strong as observed for intron 1 (**Fig 5B**). Similarly, the splicing efficiencies of *PAC1* intron 5 and intron

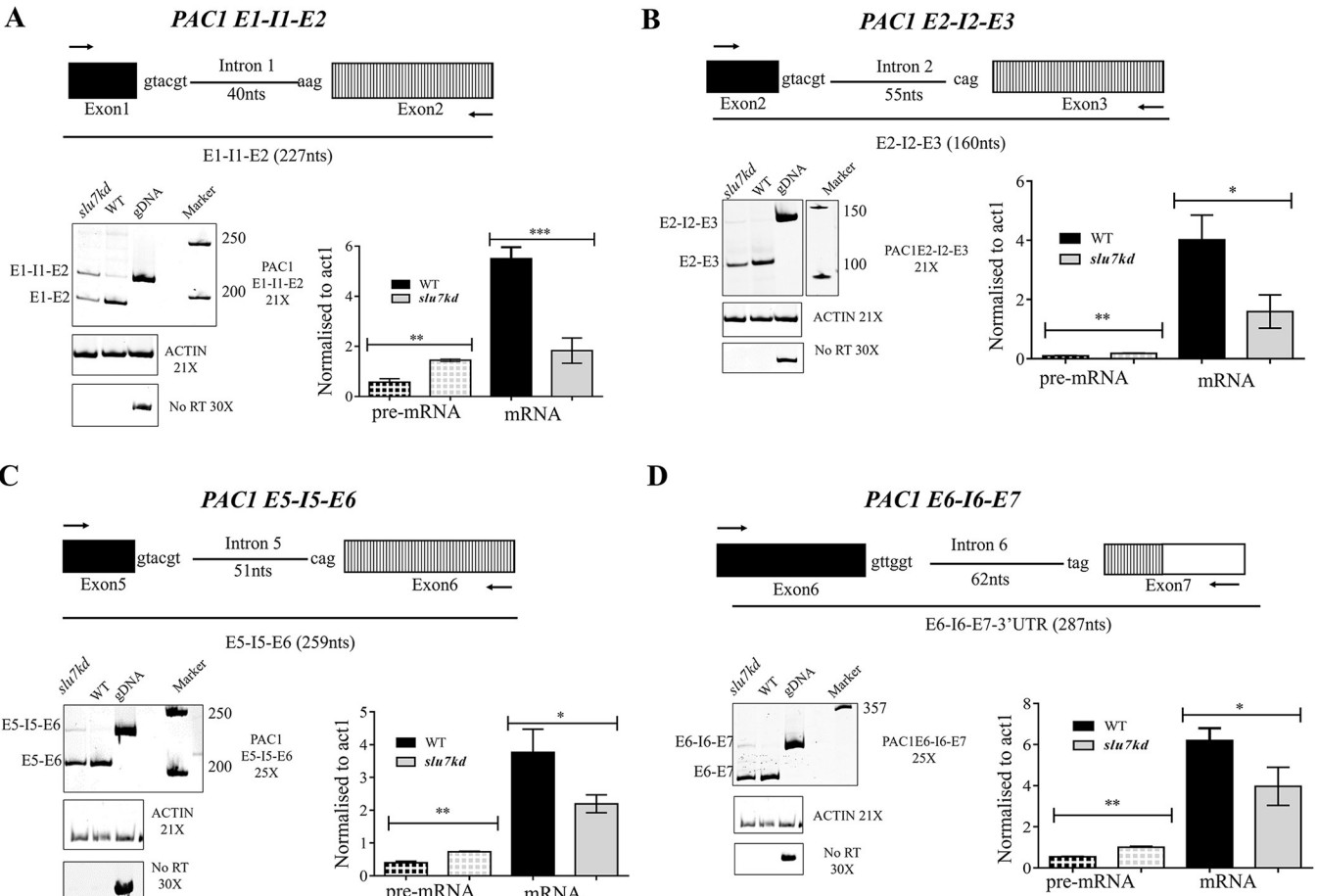

**Fig 5. Knockdown of Slu7 results in inefficient splicing of PAC1, leading to low mRNA levels.** Schematic representations show each intron together with its flanking exons. Intron length is given within brackets. RNA from WT and *slu7kd* cells grown at 30°C for 12 hours was reverse transcribed, and the cDNA was taken for limiting cycle, semi-quantitative RT-PCR using the flanking exonic primers. For each intron, the pre-mRNA (P) or mRNA (M) levels were normalized to that of the act1+ mRNA. The normalized pre-mRNA or mRNA levels are plotted. The data represent mean ± SD for three independent biological replicates. p values were determined by unpaired Student's t-test. ns, non-significant change with p > 0.05. (**A**) The splicing status of the *PAC1* intron 1 in wildtype and *slu7kd*. (**B**) The splicing status of the *PAC1* intron 2 in wildtype and *slu7kd*. (**C**) The splicing status of the *PAC1* intron 5 in wildtype and *slu7kd*. (**D**) The splicing status of the *PAC1* intron 6 in wildtype and *slu7kd*.

6, were also marginally affected (**Fig 5C and 5D**). Although exon3-exon4 and exon4-exon5 spliced mRNAs are reduced for both splice junctions, unspliced pre-mRNA was undetectable (**S6A and S6B Fig**). The failure to accumulate high-levels of unspliced pre-mRNA could be attributed to the possibility of its rapid turnover by degradation, however, a decrease in spliced mRNA levels was observed for all the splice junctions, indicating an overall decrease in mature *PAC1* transcript levels in cells depleted of Slu7. These results show that depletion of Slu7 affects the splicing of multiple introns of *PAC1*, thereby resulting in low expression of *PAC1* transcript.

To confirm that cell cycle phenotypes and splicing defects are due to the direct consequence of the conditional knockdown of Slu7, we generated a strain to express *SLU7FL* from a heterologous safe haven locus [41]. The growth arrest effects of *slu7kd* seen when grown in non-permissive YPD media were rescued when *SLU7FL* expressed from -500 bp native promoter was integrated at the heterologous safe haven locus (**S6C Fig**, compare row 2 (*slu7kd*) to row 3 (*slu7kd* expressing *SLU7FL* from safe haven locus). We also assessed the splicing efficiency of

the *PAC1 intron 1* in this growth complemented strain. These semi-quantitative RT-PCR showed *PAC1* mRNA levels were reverted to that seen in the wildtype strain (**S6D Fig**, compare lane 2 (*slu7kd*) to lane 3 (*slu7kd* expressing *SLU7FL* from safe haven locus). Taking these results together, we infer that poor splicing of *PAC1* introns triggered by Slu7 depletion is one of the contributing limiting factors for G2/M arrest arising from the defects in nuclear migration and spindle position.

## Overexpression of intronless *PAC1* from heterologous safe haven locus rescues the nuclear misposition defect caused by Slu7 depletion

ScPac1 regulates the targeting of dynein to microtubules, which is a critical determinant of nucleus positioning during mitosis [11,13]. Budding yeast *pac1Δ* cells are viable but show loss of dynein puncta in both mother and daughter cell and fail to display microtubule (MT) sliding, resulting in improper chromosome segregation. More recently, the role of its homolog in *U. maydis*, LIS1 was studied, and this report identified its essentiality for viability, cell morphogenesis, and nuclear migration [42]. Taking these reports and our findings of compromised *PAC1* splicing and its lowered expression in *slu7kd*, we hypothesized that expression from intronless *PAC1* minigene could rescue the nuclear position defect in *slu7kd* cells.

A plasmid with the intronless *PAC1* minigene (cDNA + 3'UTR) expressed from the H3 promoter and with N terminal GFP fusion was generated, taken for integration into the heterologous safe haven locus (see Materials and Methods), and subsequent expression of *PAC1* from this intronless minigene (**Fig 6A**). The overexpression of *PAC1* from the integrated minigene at the safe haven (SH) locus was confirmed by semi-RT PCR (**Fig 6B**). *slu7kd* cells with integrated *PAC1* minigene at the SH locus showed growth arrest in YPD media replicating the phenotype of the parent *slu7kd*, implying *PAC1* overexpression does not rescue the slow growth phenotype (**Fig 6C**). We performed live imaging of nuclei in cells of the *slu7kd* GFP-H4 SH:H3p-PAC1 minigene strain after 6 hrs of growth in non-permissive YPD media and compared the nuclear dynamics with that of the parent *slu7kd* GFP-H4 strain. While ~40% of *slu7kd* cells displayed nuclear division abnormally at the bud neck, *slu7kd* cells with *PAC1* minigene at SH locus had partial rescue as only 15% of cells showed nuclear division at abnormal position (**Fig 6D**, pink bar). Also noted is the concomitant increase in the percentage of cells where nuclear division occurred in the bud (**Fig 6D**, green bar). Flow cytometry analysis showed that the *slu7kd* GFP-H4 SH:H3p-PAC1 minigene strain showed a similar proportion of cells in G2/M cells as parent *slu7kd* when grown in non-permissive media for 12 hrs though the nuclear position phenotype was rescued (**S7A Fig**). Based on the two observations of G2/M phase arrest by FACS and rescue of nuclear position in *slu7kd* GFP-H4 SH:H3p-PAC1 cells, we hypothesized the potential for arrest post-nuclear division either in late anaphase or during cytokinesis in these cells. In *slu7kd* strain, we observed majority (~ 32%) of the large budded cells arrested pre–mitosis (unsegregated nuclei in mother cell, **S7B Fig**, yellow bar). However, in *slu7kd* with PAC1 minigene at the SH, a high proportion of cells (50%) were in the post-mitotic phase (segregated nuclei in mother and daughter cell, **S7B Fig**, red bar). Therefore, in *slu7kd* GFP-H4 SH:H3p-PAC1 minigene cells, and in wildtype and *slu7kd* cells taken as controls, we assessed the completion of the mitosis. We observed that *slu7kd* GFP-H4 SH:H3p-PAC1 minigene cells have segregated nuclei but were arrested post-anaphase with enhanced calcofluor staining in the neck region between the mother cell and daughter bud (**S7D Fig**). Revisiting our transcriptomic data, we found that genes involved in mitosis exit network and cytokinesis, such as *DFB1*, *MOB2*, *CDC12*, *BUD4*, and *CHS2*, were deregulated in *slu7kd* when compared to wildtype. By qRT-PCR analysis, we reaffirmed that *MOB2*, *CDC12*, and *DFB1* were expressed at higher levels in *slu7kd* when compared to wildtype

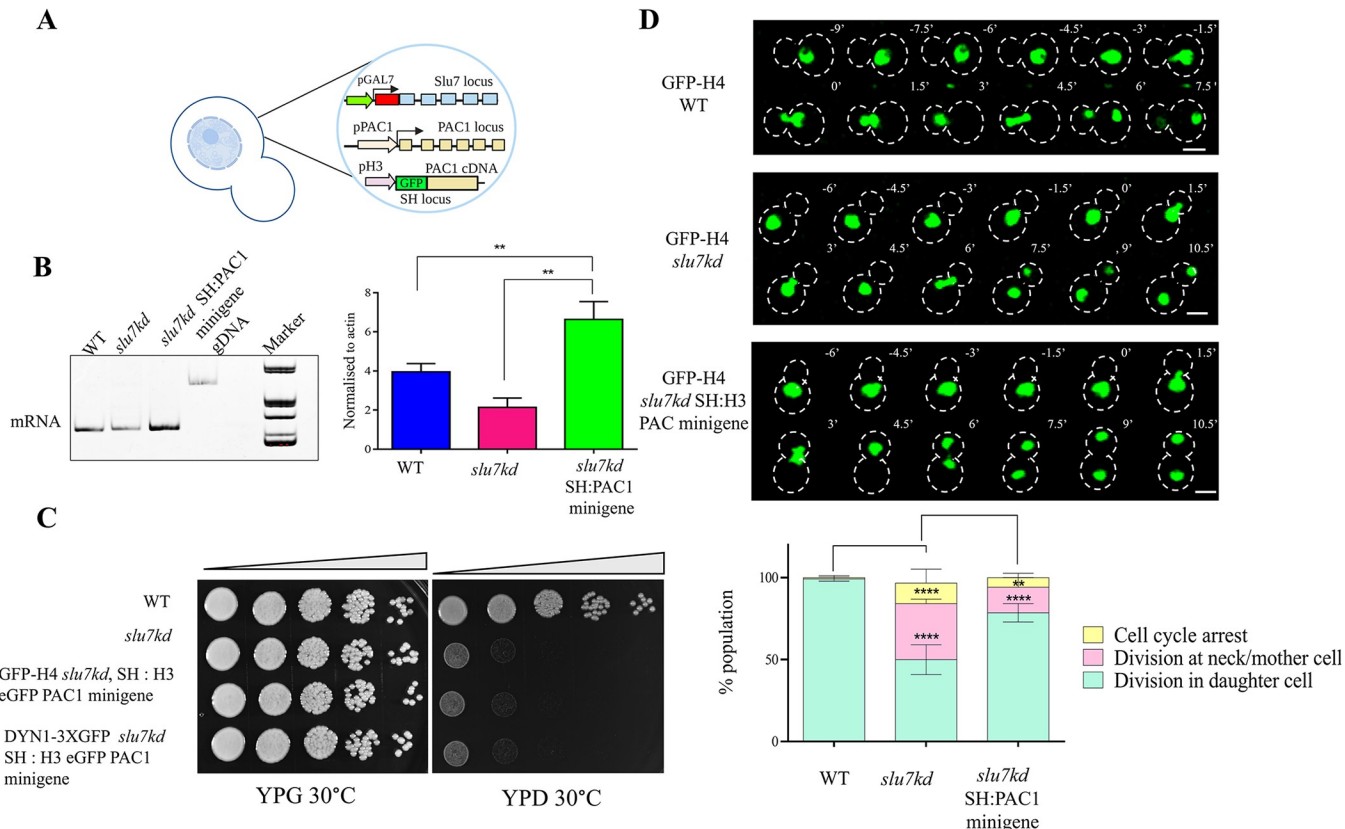

**Fig 6. Intronless *PAC1* rescues the nuclear position defect but not the growth defect in Slu7 knockdown.** (**A**) Schematic representation of *slu7kd* overexpressing intronless *PAC1* from heterologous safe haven locus. (**B**) qRT-PCR to detect mRNA levels in wildtype, *slu7kd*, and *slu7kd* expressing intronless *PAC1* minigene from safe haven loci. RNA from WT, *slu7kd*, and *slu7kd* cells expressing intronless *PAC1* grown at 30˚C for 12 hours was taken for limiting cycle, qRT-PCR using exonic primers. The experiment was done in three independent biological replicates. (**C**) Serial 10-fold dilution of 2 X 10⁵ cells from wildtype, *slu7kd* strain and *slu7kd* expressing *PAC1* minigene from safe haven, spotted on non-permissive media and monitored for growth at 30˚C for 5 days. (**D**) Time-lapse snapshots of wildtype, *slu7kd*, and *slu7kd* overexpressing intronless *PAC1* minigene strains to visualize nuclear (GFP-H4) dynamics after growth in non-permissive media for 6 hours. T = 0 was taken when nuclei approached the mother and daughter cell neck region. Bar, 5μm. Quantifying defects in the nuclear migration in *slu7kd*, and *slu7kd* with integrated with intronless *PAC1* minigene at safe haven locus (SH), both with histone GFP-H4 marked nuclei. Data were also compared to nuclear migration in the wildtype GFP-H4 strain. Percentages of cells showing normal pattern of nuclear migration into daughter bud in large budded cells followed by nuclear division, of cells with nuclear migration defect and division at the neck between the mother and daughter cell (pink), and of cells with no nuclear migration are plotted yellow. The data represent mean ± SD for three independent biological replicates with 50 cells each. One-way ANOVA test followed by Turkey's multiple comparison test was used to calculate the statistical significance of differences between the population (the p values show the difference compared to wildtype.

(**S7D Fig**). These data together suggest that multiple players involved in cell cycle progression are affected in *slu7kd* cells in *C. neoformans*.

## Discussion

Several lines of evidence in model systems established a possible link between splicing and cell cycle. The earliest study to link splicing factors to cell cycle was the *S. cerevisiae cdc40* mutant which exhibited a delay in G1/S transition and growth arrest in non-permissive temperature [24]. CDC40 was later identified as Prp17, a second step splicing factor. Several studies have demonstrated the role of splicing factors in cell cycle progression [26,30,43,44]. In this study, we identified CnSlu7 regulates nuclear migration from the mother cell to the daughter during mitosis in *C. neoformans* (**Fig 7**). Its critical role in positioning the nucleus is partly executed through efficient splicing of introns in cell cycle regulators and spindle positioning cytoskeletal

elements. Through our findings, we propose a model in which Slu7, a splicing factor, mediates the splicing and appropriate levels of *PAC1* mRNA. This in turn facilitates the role of Pac1 in targeting dynein to microtubules, followed by attachment of dynein to the daughter cell cortex. These coordinated events, with fine-tuned levels of dynein clustering, could generate the pulling force to move the nucleus into the daughter cell. Our findings uncover a Slu7-dependent cell cycle mitotic progression mechanism in *C. neoformans*.

In *S. cerevisiae* and *S. pombe*, the role of Slu7 in cell cycle is unknown. Studies in human cells siRNA-mediated knockdown of Slu7 triggered mitotic delay and defects in centriole duplication [45,46]. A recent study in human cell lines has demonstrated the role of Slu7 in mitosis and genome stability through the regulation of SR protein splicing where knockdown of Slu7 resulted in DNA damage, R–loop accumulation, and loss of sister chromatid cohesion [47]. Transcriptome analysis of CnSlu7-depleted cells indicates an increased expression of DNA damage repair genes of the RAD family. In addition, a mRNA binding protein *CNAG_04848* was expressed at low levels in CnSlu7 knockdown cells when compared to wild-type; an ortholog search revealed it to be a putative SRP2 homolog in *C. neoformans*. This line of evidence, together with the recent data that Slu7 depletion of human cell lines resulted in mis–splicing of transcripts for SR protein, hints at a possible role of CnSlu7 in preventing R–loop formation and DNA damage in *C. neoformans*.

Our study revealed slower G2-M transition in Slu7 knockdown cells. Even Slu7 knockdown cells expressing *PAC1* minigene, which substantially rescued nuclear migration defects, failed to exit mitosis, suggesting roles for Slu7 in various phases during cell cycle. Indeed, genes involved in the mitotic exit network (MEN), such as homologs of *MOB1* and *DBF2*, were upregulated in *slu7kd*. Transcriptome profile of *C. neoformans* during cell cycle progression demonstrated periodic transcription of ~18% of genes [18]. Progression through the mitotic cycle requires periodic regulation of these genes, which are achieved in specific cell cycle phases, ensuring proper DNA duplication and segregation. Splicing becomes indispensable in this case since the newly transcribed pre-mRNAs must be processed to mRNA at appropriate levels for translation of the encoded proteins. Interestingly, CnSlu7 appeared to be periodically transcribed during the cell cycle, unlike many other splicing factors. This suggests Slu7 itself might be subjected to cell cycle dependent regulation and hints at consequent temporal regulation of splicing by Slu7. The relationship between periodic abundance of Slu7, cell cycle, and mitotic progression needs further investigation.

We observed nuclear position defect, spindle position defect, impaired G2 –M transition, and impaired progression after metaphase upon CnSlu7 depletion. In addition, dynein puncta are not localised in the daughter bud in *slu7kd* cells. A central role has been established for microtubule and microtubule associated proteins in coordinated regulation of nuclear movement in fungi [10]. A series of experiments in *S. cerevisiae* indicate that dynein is targeted to the plus end of the microtubule through Pac1, and cortical attachment of dynein followed by cytoplasmic microtubule sliding against the cortex powers the nuclear migration [11,13–15]. In *S. cerevisiae pac1Δ* cells, cytoplasmic localization of dynein was not observed. A similar phenotype was observed in CnSlu7 depleted cells, and *PAC1* mRNA levels were low when compared to wildtype. Therefore, the impaired expression of *PAC1* can explain the nuclear positioning defect triggered by Slu7 knockdown in *C. neoformans*. Further, we delineated a molecular mechanism demonstrating that Slu7 is essential for efficient splicing of specific introns in *PAC1*, and that expression of intronless *PAC1* partially rescues the nuclear position defect seen in Slu7 knockdown cells. Live imaging of dynein was not feasible and hence we adopted fixed cell imaging after 12 hrs of culture in non–permissive media where we observed loss of dynein puncta in Slu7 depleted cells. Imaging Dyn1-GFP in *slu7kd* cells expressing *PAC1* minigene was technically challenging since in this particular strain, a high proportion of

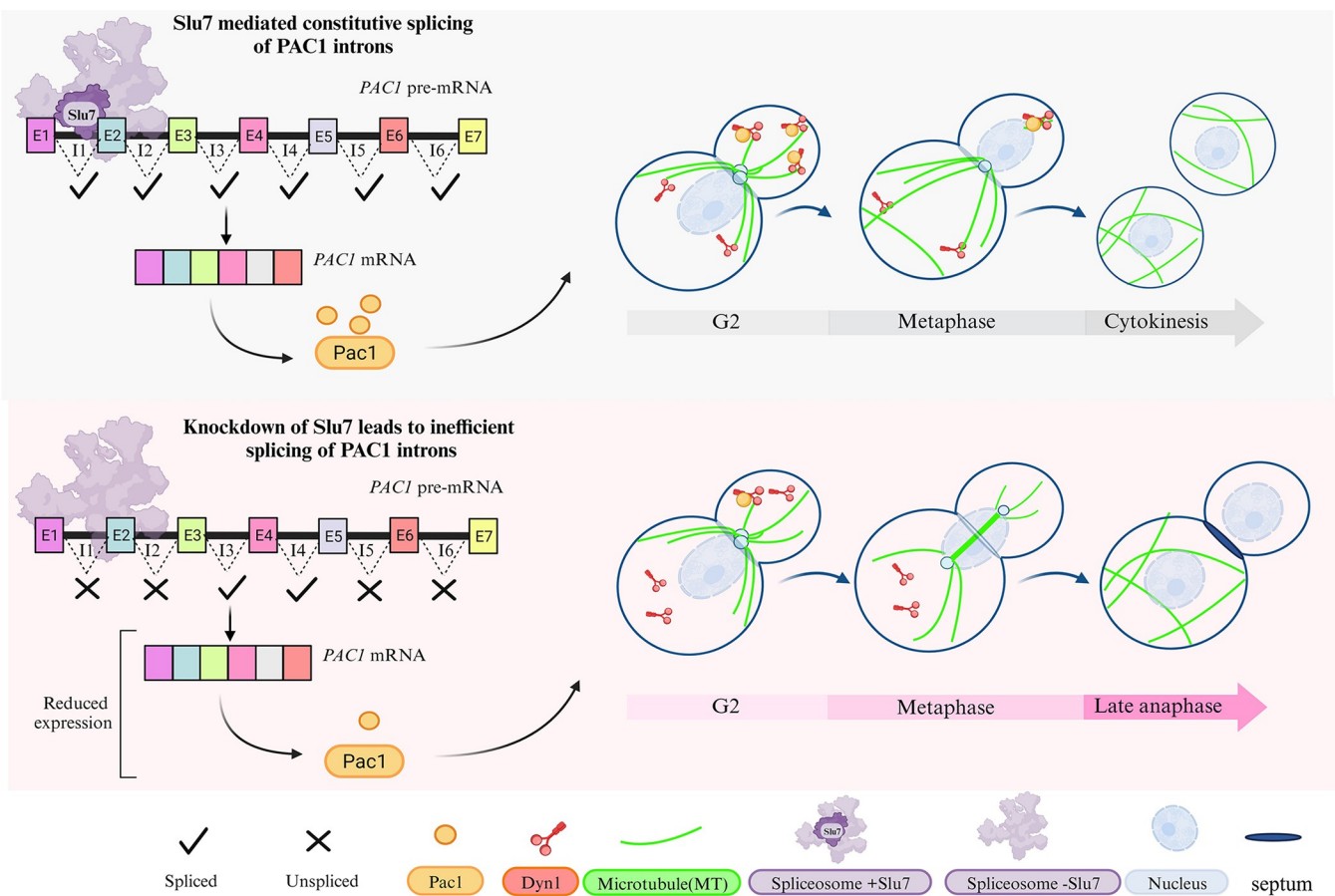

**Fig 7. A model describing the molecular pathway by which Slu7 regulates the progression of cell cycle in *Cryptococcus neoformans*.**

cells arrested post nuclear segregation, where the dynein signal is redistributed in the cytoplasm and no longer appears as puncta (**S7C Fig**).

*In vivo* studies in *S. cerevisiae* have shown that Dyn1, Pac1, and Bik1 interact at the plus end of the microtubule and Pac1 overexpression enhances the frequency of cortical targeting for dynein and dynactin without affecting the stoichiometry of the complexes at the plus end [13]. In addition, studies from *S.cerevisiae* showed Pac1, Dyn1, and Bim1 are involved in nuclear migration to the neck; however, the deletion of *BIM1* does not affect Pac1 localization to microtubules [11]. Recent studies in *C. neoformans* have shown that the spatial distribution of dynein and Bim1 plays an essential role in nuclear migration [17]. As the cell cycle progresses, a localized dynein patch is formed in the daughter cortex along the axis of symmetry, which pulls the nucleus inside the daughter bud [16]. In our transcriptomic data, we observe that *BIM1* transcript was at higher levels in *slu7kd* when compared wildtype while *DYN1* transcript levels were unaffected. In this context, any roles of Slu7 in regulating the dynamics of cytoplasmic microtubules or associated cortical proteins such as Bim1 can be addressed in the future. This also raises interesting questions regarding the functional conservation of Dyn1 and Pac1 interaction and recruitment to microtubules in *C. neoformans*. On this note, simultaneous tagging of *DYN1* and *PAC1* in the background of MT-tagged Slu7 knockdown is technically challenging because of the limited availability of markers in *C. neoformans*.

Though Slu7 is studied and explored in model organisms, past and current studies highlight its role in the cell cycle, DNA methylation, and its canonical role as a splicing factor [48]. A recent study in *C. neoformans* has shown that many spliceosomal proteins are conserved between humans and fungi, enabling the splicing of introns with degenerate features while simultaneously ensuring splicing fidelity [49]. *C. neoformans* is particularly interesting for investigations on splicing and splicing-related processes because it harbours more than 40,000 introns, and introns have been reported to regulate gene expression [34]. However, the *in vivo* roles of most of the essential splicing factor of *C. neoformans* have not been established yet. Here, we report splicing dependent role of CnSlu7 in regulating the nuclear migration from the mother cell to the daughter bud during mitosis in *C. neoformans*. Overall, our findings suggest the diverse functions of Slu7 in regulating gene expression for cell cycle regulators and cytoskeletal components, whose combined effects ensure timely cell cycle transitions and nuclear division during mitosis. In the future, combining data from different experiments such as IP-MS, expression profiles of splicing factors in different cell cycle phases, and post-translational modifications of splicing factors can pave the better way to understand the regulation of *C. neoformans* cell cycle progression by splicing.

## Materials and methods

### Yeast strains, primers, and media used in this study

The strains and primers used in this study are listed in the **S1** and **S2** **Tables**, respectively.

### Construction of Slu7 conditional knockdown

The conditional knockdown of Slu7 was generated by replacing the endogenous promoter of Slu7 with the GAL7 promoter [50]. The 5'UTR– 1KB (Fragment 1) of the start codon was PCR amplified from the H99 genomic DNA and cloned into pBSKS+ vector using the oligos listed in **S2 Table**. Similarly, 1KB (Fragment 2), including the start codon, was PCR amplified and cloned into pBSKS+ vector. After confirmation of insert by sequencing, the two fragments were subcloned into pGAL7 mCherry vector with hygromycin selection marker, fragment 1 at Sac1 site and fragment 2 at SalI and ApaI, respectively to obtain the final clone where the reading frame of Slu7 was N terminally fused with mCherry. The resulting plasmid was partially digested with Sac1 and ApaI (the other SacI at the 3'end of fragment1 was deleted by end filling) was introduced into wildtype H99 (a kind gift from Prof. Kaustuv Sanyal, [51]) by biolistic transformation. Hygromycin positive transformants were confirmed by PCR using the oligos listed in **S2 Table**.

### Construction of Slu7 knockdown in the background of GFP-H4, GFP-CENPA, GFP-TUB1, DYN1-GFP and mad2Δ-GFP-H4

To visualize the dynamics of the nucleus, kinetochore, microtubules, and dynein, Slu7 conditional knockdown was generated in the background of GFP-H4 (a kind gift from Prof. Kaustuv Sanyal, [6]), GFP-CENPA [17], GFP-TUB1 [52], and DYN1-GFP [16] and mad2Δ-GFP-H4 [53] strains. pGAL7 mCherry vector containing Slu7 F1 and F2 fragments was digested with SacI and ApaI, and the fragment was introduced into the above strains by biolistic transformation. The transformants were selected by hygromycin resistance and confirmed by PCR using the primers listed in **S2 Table**. The growth kinetics of the generated strains were compared to the parent Slu7 knockdown by 10-fold serial dilutions of broth cultures grown to equivalent $OD_{600}$.

## Construction of SLU7 FL and PAC1 minigene complementation strain

Complementation of SLU7FL was achieved by PCR amplification of SLU7 fragment along with promoter and terminator from the endogenous loci using the primers listed in **S2 Table**. The amplified 2.2KB fragment was cloned into E.coRV site of pBSKS vector and subsequently cloned into PstI and SalI site of pSMDA25 vector [41]. The resultant plasmid was linearised using BaeI and transformed into the parent Slu7 knockdown strain, and the transformants were selected for nourseothricin resistance for integration at the safe haven locus. As a negative control, an empty vector linearized with BaeI was transformed into *slu7kd* GFP-H4 cells. The positive transformants were confirmed by safe haven locus-specific PCR with primers listed in **S2 Table**. The downstream experiments were done in two independent transformants. For PAC1 minigene complementation in *slu7kd*, PAC1 cDNA was amplified by RT PCR from wildtype H99 cDNA pool generated by reverse transcription using the primers listed in **S2 Table**. The cDNA fragment and the 3'UTR fragment amplified from gDNA were combined into a single fragment by overlap PCR. This insert was cloned into SpeI site of PVY7 vector containing H3 promoter and eGFP/mCh tag at N terminal. The insert was sequence verified and subsequently taken as XbaI-XmaI fragment and cloned into SpeI-XmaI sites in pSMDA57/25 vector [41]. The resultant plasmid was linearised using BaeI and transformed into parent Slu7 knockdown strain with GFP-H4 and DYN1-GFP, and the transformants were selected for kanamycin resistance (for GFP-H4 strain) and nourseothricin resistance (for DYN1-GFP strain) for integration at the safe haven locus. The positive transformants were confirmed by safe haven locus-specific PCR with primers listed in **S2 Table**.

## Media and growth conditions

All strains were initially grown in permissive media YPG (2% peptone, 2% galactose, 1% yeast extract) overnight and then subcultured into either permissive media or non-permissive media YPD (2% peptone, 2% glucose, 1% yeast extract) and grown for 6 hrs/12 hrs depending on the experiments. All strains were grown at 30˚C.

## Live cell imaging

The conditional knockdown strain of Slu7 in the background of GFP-H4, GFP-CENPA, GFP-TUB1, and mad2Δ GFP-H4 was grown in permissive media overnight and reinoculated into non-permissive media for 6 hrs. The cells were pelleted and washed with PBS, resuspended in YPD, and placed on a slide containing 2% agarose patch made of YPD. After the cells were settled, the agarose patch was then transferred to the confocal dish and proceeded for live imaging. Live imaging was performed at 30˚C in Leica SP8 upright microscope with argon laser, HC PL APO CS2 40x/1.30 oil objective, and Tokai hit stage top temperature-controlled chamber. Images were taken sequentially by switching between the GFP filter line (488 nm of emission and excitation wavelength of 580nm) and DIC. The images were collected for every 30 second interval, with 10% laser intensity and 0.5μm stacks, for 90 minutes, respectively, except for GFP-TUB1, where the images were collected every 30 second interval, with 10% laser intensity and with 0.3μm stacks. As a control, wildtype grown on the same day was imaged with the exact setting. *slu7kd* GFP-H4 with PAC1 cDNA complemented and imaged in the same condition along with wildtype and parent *slu7kd* as controls. For GFP-TUB1 strains, a representative image (**Fig 3A**) was collected from LSM 880 AxioObserver, PL APO 100X 1.40 oil objective, laser power of 5.5% and GFP emission and excitation of 488nm and 546nm, respectively, 0.3μm stacks. For the quantification of GFP-TUB1, images taken on Leica SP8 were used.

## Fixed cell imaging and processing

The conditional knockdown of Slu7 in the background of GFP-H4 and wildtype H99 were grown in permissive media overnight and then shifted to non-permissive media. The sample was collected every 2 hrs till 12 hrs of growth to observe the loss of Slu7-mCherry signal after the transfer into non-permissive media. The cells were pelleted down, washed with PBS, and placed on an agarose patch contained in a microscopic slide. The coverslip was placed on the patch and was proceeded for imaging. The images were acquired in confocal laser scanning microscopy (Zeiss LSM 880) using a 63X oil immersion objective lens, and the images were processed by Zen Blue software and then analyzed by ImageJ. For budding index calculation, *slu7kd* GFP-H4 and wildtype H99 grown in non-permissive media for 12 hrs were imaged in the same system and processed by ImageJ. The diameter of the mother and the daughter cell was measured along the axis using the line tool in ImageJ. The budding index was calculated as the ratio of the diameter of the daughter cell to that of the mother cell.

For calcofluor white (CFW) staining to visualize the cell wall, wildtype, *slu7kd*, and *slu7kd* expressing intronless PAC1 transcript from safe haven was grown in non-permissive media for 12 hrs, was fixed with paraformaldehyde and stained with calcofluor white (20μg/ml) for 5 mins. Before imaging, an equal concentration of KOH was added to the cells. The cells were placed on an agarose patch on a microscopic slide. The cover slip was added, the cells were imaged in Zeiss Leica spinning disk confocal microscope using a 100X oil immersion objective lens, and the images were processed and analyzed by ImageJ. Exposure time and laser settings for Green Fluorescent Protein (GFP) channel were 0.8 s and 30% with a step size of 0.5 μm, and for CFW, the exposure time was 0.15 and setting of 2%, respectively.

To visualize Dyn1-GFP puncta in *slu7kd* and wildtype were initially grown in permissive media, transferred into non-permissive media, and allowed to grow for 12 hrs. The cells were pelleted, washed with PBS, mounted on a microscopic slide with a coverslip, and taken for imaging. Imaging was carried out at room temperature using a DeltaVision RT microscope with an Olympus 100X, oil-immersion 1.516 NA objective. Exposure time and transmittance settings for Green Fluorescent Protein (GFP) channel were 0.2 s and 100%, respectively, and the images were automatically deconvoluted using softWoRx 6.1.3, the software used by Delta-Vision, and the presence or absence of puncta was analyzed using ImageJ. The brightness and contrast of the images were adjusted using ImageJ and applied across the entire image for both wildtype and *slu7kd*.

## Synchronization and flow cytometry

Wildtype and *slu7kd* cells were grown in permissive media overnight and shifted to non-permissive media. An aliquot of cells was collected every 2 hrs and analyzed by flow cytometry. For early S phase growth arrest, the knockdown and wildtype were grown in permissive media was transferred to non-permissive media for 2 hrs to deplete the overexpressed Slu7, and then hydroxyurea (15 mg/ml) was added to the cultures. After 4 h of growth, the synchronized cells were pelleted down, washed with pre-warmed YPD media, and resuspended into fresh YPD media. Aliquots were collected every 2 hrs of release into YPD till 8 hrs. The cells were fixed with 70% ethanol and digested with RNase overnight. The cells were stained with PI (1μg/ml) and proceeded for flow cytometry.

Additionally, cell cycle analysis was carried out by adding a sublethal concentration of thia-bendazole to the synchronized population during release into YPD media. First, the sublethal concentration of TBZ was determined by dilution spotting of knockdown in YPD plate containing 2, 4, 6μg/ml TBZ. Since Slu7 knockdown showed sensitivity to TBZ at 4μg/ml, this concentration was fixed for the flow cytometry experiment. The HU synchronized cells were

released into YPD media containing 4μg/ml of TBZ, and aliquots of cells were collected every 2 hrs till the 8$^{th}$ hour and subjected to cell cycle analysis. Cell cycle analysis was done using BD FACS verse of 10000/50000 events. The files were analyzed using Flowjo software to obtain the percentage of cells in the G1/S and G2/M phases of the cell cycle.

## RNA seq analysis

According to manufacturer protocol, RNA was isolated from three independent batches of conditional knockdown of Slu7 and wildtype H99 cells by trizol reagent. The RNA samples were purified by treating with DNaseI (NEB) to remove the genomic DNA contamination. The purified RNA samples were sent for deep transcriptome sequencing followed by analysis to Agri Genome Labs Pvt. Ltd., Kochi, India. In brief, the cDNA sequencing libraries were generated using TruSeq Stranded Total RNA with Ribozero gold following the manufacturer's protocol. The libraries were sequenced on Illumina HiSeq X10 platform, providing paired end reads. The raw reads retrieved from the sequencer were carried forward for low quality reads filtration (Q < 30), and the cleaned reads were aligned to the Cryptococcus reference genome using HISAT2 v2.2.1 with default parameters. The expression levels of genes between Slu7 conditional knockdown and wildtype were measured as fragments per kilobase of exon per million fragments mapped (FPKM) using the Cufflinks v2.1.1 package. Differentially expressed genes were selected based on FDR corrected p-value < 0.05 and absolute log2(fold change) > 1 using cuffdiff.

## Semi quantitative RT-PCR and qRT-PCR

Reverse transcription was done using DNase-treated RNA using MMLV RT (NEB) according to the manufacture protocol. 100–200 ng of cDNA was used to carry a limiting amount of PCR reactions to amplify the spliced and unspliced products. The PCR products were resolved on 8% native polyacrylamide gels. Signal intensities for the products were obtained by staining the gel and measuring the intensities using multigauge (version 2) and normalized to Actin control. For qRT-PCR, reverse transcription was carried out using 10μM oligo dT primer and MMLV RT (NEB) according to the manufacture protocol. qRT-PCR reactions were set up with 20-30ng of cDNA, 250 nM gene-specific primers, and FastStart Universal Sybr Green Master mix (Merck) in CFX Opus real-time system (BioRad). Fold change in transcript levels for deregulated genes was calculated as the difference in cycle threshold value between Slu7 knockdown and wild type. To obtain the normalized threshold value (ΔΔCt), the ΔCt value was calculated by subtracting the Ct value for internal control–Actin, from the Ct value for each gene of interest. Then ΔΔCt was calculated by subtracting the wild type ΔCt value from the ΔCt value obtained for Slu7 knockdown. The fold change was calculated as 2^-(ΔΔCt). Primers used, and their sequences are given in **S2 Table**.

## Western blot

Wildtype and Slu7 conditional knockdown cells were grown in permissive and non-permissive media for 12 hrs. The cells were pelleted and washed with PBS. This was followed by vortexing the cell pellet with acid-washed glass beads (Sigma, Cat. No. G8772) with 13% TCA for 30 mins at room temperature. The lysate was separated from the beads and pelleted at 13,000 rpm for 10 minutes. This was followed by 80% acetone wash to remove the residual TCA. The pellet was air-dried and resuspended in 4X Lamelli buffer (0.02% Bromophenol blue, 30% glycerol, 10% SDS, 250 mM Tris-Cl pH 6.8, 5% β-mercaptoethanol) and denatured at 95˚C for 5 mins. The samples were loaded on 10% SDS PAGE, followed by electrophoresis, and transferred into PVDF membrane for 1 hrs at 16 V using the semi-dry method (BioRad). The membrane was

blocked with 3% BSA / 5% skimmed milk in TBS, depending on the primary antibody. After blocking, the membrane was incubated with primary antibody overnight at 4˚C. The membrane was washed thrice with TBST (1X TBS + 0.1% tween20) and incubated with secondary antibody for 1 hour at room temperature. The blot was washed thrice with TBST, and then the signals were then detected using the chemiluminescence method (Millipore Immobilon Forte HRP substrate).

Primary antibodies used for western blot analysis were mouse anti-GFP (dilution 1:20000) (Thermofisher Scientific, Cat. No. A11122) and rabbit anti- mCherry (dilution 1:10000) (Abcam, Cat. No. ab213511). Secondary antibodies used are goat anti-rabbit HRP conjugated antibodies (dilution 1:10,000) (BioRad Cat. No. 1706515) and goat anti-mouse HRP (BioRad Cat. No. 1706516).

## Phylogenetic tree, volcano plot, and heatmaps

The phylogenetic tree in **Fig 1A** was generated by Meta software (version 11). Slu7 protein sequences were collected for each organism, followed by multiple sequence alignment using Clustal W, and maximum likelihood algorithm was applied to generate the phylogenetic tree. The multiple sequence alignment in **Fig 1A** was done using Clustal W and visualised with Jalview (version 2.11.3). The volcano plot in **Fig 5A** was generated using R studio (version 4.2.1) using the ggplot2 package. The heatmaps in **Fig 4B** were generated using R studio (version 4.2.1) using the pheatmap package. The FPKM values are mentioned inside each cell, and the scale denotes the $\log_{10}$(FPKM) values of wildtype and knockdown, respectively.

## Statistical tests

All plots mention the standard deviation and the mean of three independent experiments. As mentioned in the figure legends, the statistical significance of differences was calculated with one-way ANOVA with Tukey's multiple comparison tests or unpaired t-test. P values $\leq 0.05$ were considered significant. All analyses were done using GraphPad prism version 6.01. Flow cytometry and western blot images shown in the figures are representatives of at least three independent biological replicates which showed similar results. The significances are mentioned as ns, $P \geq 0.05$; *—$P < 0.05$; **—$P < 0.01$; ***—$P < 0.001$; ****—$P < 0.0001$.

## Supporting information

**S1 Fig. CnSlu7 is evolutionary close to higher eukaryotes than the other fungal orthologs and its depletion results in slow growth.** (**A**) Phylogenetic tree conservation of Slu7 protein among fungi group. The tree was generated using MEGA 11 with Maximum Likelihood method. The branch lengths measured in the number of substitutions per site is denoted above the branches. (**B**) Schematic representation of Slu7 knockdown strain. (**C**) Quantification of RNA depletion in *slu7kd* and wildtype cells after the shift into non-permissive media for 6 hours and 12 hours by qRT PCR. (**D**) Western blot to detect mCherry tagged Slu7 protein in *slu7kd* grown in non-permissive media for 6 hours and 12 hours at 30˚C. The blot was probed with anti-mCherry as described in the Materials and Method section. pSTAIRE was used as loading control. Knockdown strain grown in permissive media for 12 hours was used as positive control.
(TIF)

**S2 Fig. Conditional knockdown of Slu7 shows splicing defect of *TFIIA intron 1* and slower progression in mitosis.** (**A**) Splicing defect of TFIIA intron 1 in *slu7kd* and wildtype grown in YPD for 12 hours. The reactions marked as "No RT 29 X" denote semi- quantitative PCR

performed on DNase treated RNA sample, without reverse transcription of RNA to cDNA. (**B**) Growth profile of broth cultures of wildtype and *slu7kd* grown in non-permissive media. The data represent mean ± SD for three independent biological replicates. (**C**) Flow cytometry analysis of synchronous wildtype and Slu7 conditional knockdown after release into non-permissive media. The 0[th] time point indicates the HU arrested cell population before the release into permissive media. The percentage at the top represents the % of cells in the G2/M phase at the end of 6 hours, N = 3. (**D**) Serial 10-fold dilution of 2 X 10$^5$ cells from wildtype, *slu7kd* strain. The strains were grown in non–permissive media for 6 hours and 12 hours and spotted on permissive media and monitored for growth at 30˚C for 5 days to assess the loss of viability. (TIF)

**S3 Fig. Dynamics of nucleus and kinetochore patterning in *slu7kd*.** (**A**) Serial 10-fold dilution of 2 X 10$^5$ cells from wildtype, *slu7kd* strain, each of which had marked reporters for monitoring mitosis. These strains with GFP-H4, GFP-TUB1, and GFP-CENPA reporters were spotted on non-permissive media and monitored for growth at 30˚C for 5 days. (**B**) Time-lapse snapshots of *slu7kd* and wildtype cells with GFP-H4 reporter to visualize nuclear dynamics after growth in non-permissive media for 12 hours. T = 0 was taken when the nucleus enters the neck region in the wildtype panel. In the knockdown panel, the timestamps are mentioned right from the start of the imaging. Bar, 5μm. (**C**) Time-lapse snapshots of *slu7kd* and wildtype cells with GFP-CENPA reporter to visualize the kinetochore dynamics after growth in non-permissive media for 6 hours. T = 0 represents the start of the live imaging. Bar, 5μm. (TIF)

**S4 Fig. Slu7 mediated defects are not under the surveillance of Spindle Assembly Checkpoint (SAC).** (**A**) Serial 10-fold dilution of 2 X 10$^5$ cells from wildtype and *slu7kd* spotted on non-permissive media containing 2μg/ml and 4μg/ml thiabendazole and monitored for growth at 30˚C for 5 days. Flow cytometry analysis of cells from wildtype and *slu7kd* strain withdrawn at various time points after inoculation of HU- synchronised cells into non-permissive media containing 4μg/ml thiabendazole. The percentage figures given at the top represents the % of cells in the G2/M phase at the end of 6 hours, N = 3. (**B**) Serial 10-fold dilution of 2 X 10$^5$ cells from wildtype and *slu7kd* in the background of *mad2Δ* GFP-H4 spotted on non-permissive media and monitored for growth at 30˚C for 5 days. (**C**) Flow cytometry analysis of cells from wildtype and *slu7kd* strain in the background of *mad2Δ* GFP-H4 withdrawn at various time points after inoculation of HU- synchronised cells into non-permissive media. The percentage figures given at the top represents the % of cells in the G2/M phase at the end of 8 hours, N = 3. (**D**) Time-lapse snapshots of *slu7kd* and wildtype cells in the background of *mad2Δ* GFP-H4 to visualize nuclear dynamics after growth in non-permissive media for 6 hours. T = 0 was taken when the nucleus enters the neck region between the mother and daughter cell. Bar, 5μm. (TIF)

**S5 Fig. Slu7 knockdown leads to deregulation of transcripts involved in various cellular functions.** (**A**) Volcano plot representing the differential gene expression between *slu7kd* and Wildtype. The dotted line parallel to X-axis denotes the cutoff for -log$_{10}$(FDR), of 1.303 (FDR value less than 0.05), and the dotted lines parallel to Y-axis denote the cutoff for log$_2$(fold change) of –1 or +1. (**B**) The bar chart represents the upregulated and downregulated genes in Slu7 knockdown compared to wildtype. (TIF)

**S6 Fig. Intron 3 and intron 4 of PAC1 transcript are not dependent on Slu7 for splicing.** Schematic representations show each intron together with its flanking exons. Intron length is given within brackets. RNA from WT and *slu7kd* cells grown at 30˚C for 12 hours was taken for limiting cycle, semi-quantitative RT-PCR using the flanking exonic primers. For each intron, the pre-mRNA (P) or mRNA (M) levels were normalized to that of the *ACT1* mRNA. The normalized pre-mRNA or mRNA levels are plotted. The data represent mean ± SD for three independent biological replicates. p values were determined by unpaired Student's t-test. ns, non-significant change with p > 0.05. (**A**) The splicing status of the *PAC1* intron 3 in wild-type and *slu7kd*. (**B**) The splicing status of the *PAC1* intron 4 in wildtype and *slu7kd*. (**C**) Serial 10 dilutions starting from 2 X $10^5$ cells of wildtype, Slu7 conditional knockdown, and two transformants expressing Slu7FL from safe haven locus in the background of *slu7kd* were spotted on non-permissive media. The image was obtained after incubating the plates at 30˚C for 5 days. (**D**) The splicing status of the *PAC1* intron 1 in wildtype, *slu7kd*, and *slu7kd* expressing Slu7FL from safe haven locus. RNA from WT, *slu7kd*, and *slu7kd* cells expressing Slu7FL from safe haven locus grown at 30˚C for 12 hours was taken for limiting cycle, semi-quantitative RT-PCR usinssg the flanking exonic primers. The experiment was done in three independent biological replicates.
(TIF)

**S7 Fig. Slu7KD with intronless PAC1 arrest post nuclear division in late anaphase / cytokinesis.** (**A**) Flow cytometry analysis of cells from wildtype, *slu7kd* and *slu7kd* expressing PAC1 minigene grown in non-permissive media (YPD) for 12 hours. (**B**) The percentage of cells at various phases of the cell cycle, based on bud and nuclear position was measured using *slu7kd* (n = 100), wildtype (n = 100) and *slu7kd* SH::PAC1 minigene (n = 100) was measured in fixed cells (4% paraformaldehyde) after growth in YPD for 12hrs, respectively. The data represent mean ± SD for three independent biological replicates. One-way ANOVA test followed by Turkey's multiple comparison test was used to calculate the statistical significance of differences between the population (the p values show the difference compared to the wildtype vs *slu7kd*, *slu7kd* vs *slu7kd* SH:PAC1 minigene). Snapshots of wildtype GFP-H4 WT, *slu7kd* GFP-H4, and *slu7kd* with H3:PAC1intronless minigene GFP-H4 cells stained with calcofluor white to visualize the cell wall. Bar, 5μm. (**C**) Localization of Dyn1 in the wildtype, *slu7kd* and *slu7kd* with PAC1 minigene cells at different stages expressing Dyn1-3xGFP upon their growth in the non-permissive conditions. Percentages of cells with different pattern of dynein signal are quantitatively represented in the bar graph. The yellow bar represents cells with clustered dynein puncta both in mother and daughter bud in large budded cells. The green bar represents cells with multiple dynein puncta only in the mother bud of large-budded cells, and blue bar shows % cells with no dynein puncta either the mother and daughter bud of large budded cells. The data represent mean ± SD for three independent biological replicates with n ≥ 34 large-budded cells. Bar, 5μm. This experiment comes with a technical limitation to capture the nuclear position with respect to dynein position. The low sample size (metaphase cells) in *slu7kd* overexpressing *PAC1* minigene is because high proportion of cells arrested post mitosis. (**D**) qRT-PCR to assess the deregulation of cytokinesis-related genes in knockdown and wild-type cells after the shift into non-permissive media for 12 hours.
(TIF)

**S1 Table. Strains used in this study.**
(XLSX)

**S2 Table. Sequences of the primers used in this study.**
(XLSX)

**S3 Table. List of DEGs in *slu7kd vs* wildtype.**
(XLSX)

**S4 Table. Raw data numerical values underlying Figs 1–7 and S1–S7.**
(XLSX)

## Acknowledgments

We thank Prof. Kaustuv Sanyal for generously gifting the untagged H99 wildtype and the tagged strains with markers for cell cycle in wildtype background and pSTAIRE antibody. We thank Prof. Kaustuv Sanyal and his lab members especially PVS Satyadev and Dr. Vikas Yadav, JNCASR for helpful technical discussions during this study. Prof. Sanyal inputs during the preparation of the manuscript are gratefully acknowledged. Prof. Sachin Kotak is thanked for his inputs and discussions on dynein, and nuclear migration, during the course of the study. We thank Dr. Saravanan Palani for his inputs on live imaging. We thank Gayathri and Prof. Purusharth I. Rajyaguru lab, IISc, for help in imaging using the DeltaVision system. The help from staff at the Bioimaging confocal microscopy facility and at the flow cytometry facility is recorded with thanks. AgriGenome Labs Pvt. Ltd. is acknowledged RNA sequencing services and data analysis. The representative yeast figures used alongside graphs and the final model (Fig 7) were created using biorender.com.

## Author Contributions

**Conceptualization:** Vishnu Priya Krishnan, Usha Vijayraghavan.

**Data curation:** Vishnu Priya Krishnan.

**Funding acquisition:** Usha Vijayraghavan.

**Investigation:** Vishnu Priya Krishnan.

**Methodology:** Vishnu Priya Krishnan, Manendra Singh Negi, Raghavaram Peesapati, Usha Vijayraghavan.

**Project administration:** Usha Vijayraghavan.

**Resources:** Usha Vijayraghavan.

**Software:** Raghavaram Peesapati.

**Supervision:** Usha Vijayraghavan.

**Validation:** Vishnu Priya Krishnan.

**Visualization:** Vishnu Priya Krishnan.

**Writing – original draft:** Vishnu Priya Krishnan, Usha Vijayraghavan.

**Writing – review & editing:** Vishnu Priya Krishnan, Usha Vijayraghavan.

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
