## [Decision Letter · Decision Letter 0]

25 Apr 2024

Dear Dr. Vijayraghavan,

We are pleased to inform you that your manuscript entitled "*Cryptococcus neoformans* Slu7 ensures nuclear positioning during mitotic progression through RNA splicing" has been editorially accepted for publication in PLOS Genetics. Congratulations!

Yours sincerely,

Steven B. Haase

Guest Editor

PLOS Genetics

Eva Stukenbrock

Section Editor

PLOS Genetics

Comments from the reviewers (if applicable):

Reviewer's Responses to Questions

**Comments to the Authors:**

Reviewer #1: The revised manuscript has been improved by addressing sincerely all the issues I had raised in the previous review process.

Reviewer #2: The authors have made all the changes in the manuscript and figures that were suggested. Also, they have provided convincing data to answer my concerns on a previous submission. Therefore, I recommend its publication.

**Have all data underlying the figures and results presented in the manuscript been provided?**

Reviewer #1: Yes

Reviewer #2: Yes

PLOS authors have the option to publish the peer review history of their article (what does this mean?). If published, this will include your full peer review and any attached files.

Reviewer #1: No

Reviewer #2: No

**Data Deposition**

http://datadryad.org/submit?journalID=pgenetics&manu=PGENETICS-D-24-00241

**Press Queries**

---

## [Editor Report · Acceptance letter]

14 May 2024

PGENETICS-D-24-00241 

*Cryptococcus neoformans* Slu7 ensures nuclear positioning during mitotic progression through RNA splicing 

Dear Dr Vijayraghavan, 

We are pleased to inform you that your manuscript entitled "*Cryptococcus neoformans* Slu7 ensures nuclear positioning during mitotic progression through RNA splicing" has been formally accepted for publication in PLOS Genetics! Your manuscript is now with our production department and you will be notified of the publication date in due course.

With kind regards,

Zsofia Freund

PLOS Genetics

On behalf of:
